# Associations between serum uric acid and depression in U.S. adults: An analysis of the national health and nutrition examination survey 2007–2016

**Ertao Zhang**[1], **Jie Li**[2]*, **Yan Liu**[2], **Zhao Dong**[1], **Longyu Wang**[1], **Can Zhao**[1]

**1** Changzhi Medical College, Changzhi, Shanxi, China, **2** Department of endocrinology, Datong Third People's Hospital, Datong, Shanxi, China

* 2958026221@qq.com

## Abstract

Limited research has specifically explored the gender-specific differences in relation to serum uric acid and depression. Therefore, the research aims to bridge that gap by exploring the relationships between serum uric acid and depression, stratified by gender, in U.S. adults. In this study, the participants comprised individuals from five consecutive survey cycles (NHANES 2007–2008–2015–2016), with 20,804 qualified participants incorporated into the research. The assessment of depression was conducted utilizing the Patient Health Questionnaire-9 (PHQ-9), and we calculated the weighted averages (95% CIs) for continuous variables and the percentages (95% CIs) for categorical variables among the chosen covariates. A multivariable logistic regression model with weights was utilized to assess the relationships between serum uric acid categories and depression status. In male subjects, uric acid levels may be linked inversely to depression. However, Tertile 3 (≥303.3 μmol/L) was more likely to develop depression than Tertile 1 (<249.8 μmol/L) among women. The study shows that high serum uric acid level may reduce the incidence of depression in men, but in females, it is an independent factor that increases the risk of depression.

## Aims

The primary objective of this study was to investigate the gender-specific association between serum uric acid levels and depression in U.S. adults.

The findings from past studies on the associations between serum uric acid levels, depression, and gender are limited. And previous studies have demonstrated a link between serum uric acid levels and depression. However, only a small number number of these studies have specifically found the gender-specific associations between serum uric acid and depression. Consequently, our objective was to investigate the relationships between serum uric acid and depression, considering gender differences among U.S. adults, and to provide valuable reference for the monitoring and preventive health care measures of patients.

**Data availability statement:** The dataset generated during this study has been deposited in the Figshare repository. The data are publicly available at: Repository Name: Figshare DOI: 10.6084/m9.figshare.28557611.v1

**Funding:** The author(s) received no specific funding for this work.

**Competing interests:** The authors have declared that no competing interests exist.

## Introduction

Serum uric acid (SUA) is the end product resulting from the breakdown of purine nucleotides, specifically adenine and guanine; it is primarily known for its association with hyperuricemia [1]. In recent years, the prevalence rate of hyperuricemia has increased yearly with improvements of life. In 2020, the number of people with hyperuricemia in the world was about 930 million. Serum uric acid is an important antioxidant, contributing more than 60% of the total antioxidant capacity in plasma [2], therefore, it could be especially significant in depression. We hypothesized that higher serum uric acid levels would be inversely associated with depression.

Antioxidants are essential in neutralizing reactive oxygen species (ROS) and free radicals, safeguarding cells from oxidative stress (a process caused by damage to cellular structures due to decreased antioxidant activity). The brain, which consumes 20% of the body's oxygen, produces significant amounts of ROS, making it particularly susceptible to oxidative stress [3]. When it comes to serum uric acid, it exhibits dual behavior. While it can act as a pro-oxidant inside cells, a significant number of studies have revealed a positive correlation between serum uric acid and chronic diseases [4–6]. On the contrary, in the extracellular space, serum uric acid plays a crucial role with its main function being that of an antioxidant [7]. Furthermore, studies have linked elevated levels of serum uric acid with reduced rates of depression and anxiety [8,9]. It appears that gender plays a role in the relationship between serum uric acid and depression. Although evidence suggests serum uric acid is associated with depression in a negative manner, few studies have explored this relationship specifically in relation to gender. Prior studies have shown that women are more prone to depression than men, but men tend to exhibit higher levels of uric acid compared to females [10]. A study exclusively focusing on male adolescents has revealed a positive correlation between uric acid levels and the incidence of depression. Elevated SUA levels can serve as a biochemical indicator for early diagnosis of adolescent depression [11].

The level of uric acid in participants with depression has not reached a unified consensus. A study reported the control group was found to have lower levels of serum uric acid compared to patients with depression.(1) However, other studies suggest that due to increased oxidative stress, serum uric acid levels may decrease in depression [12]. This discrepancy can be attributed to several factors. Firstly, depression exhibits various clinical subtypes, and the serum uric acid levels may vary among these subtypes, which has not been considered in previous studies. Secondly, published studies on depression utilize diverse diagnostic criteria. The Patient Health Questionnaire (PHQ-9) is widely regarded as the standard diagnostic tool for depression. It is crucial to note that significant differences exist in the diagnostic criteria of depression across different classifications. The current study focuses on patients diagnosed with depression using PHQ-9 criteria upon discharge, and aims to explore uric acid levels, depression, and sexual activity can be interconnected in the United States.

## Methods and materials

### Study design and subjects

The National Center for Health Statistics (NCHS), which is managed by the National Center for Health Statistics (NCHS) of the Centers for Disease Control and Prevention, is a continuous, multi-departmental national survey of the civilian population of the United States. It adopts a complex, multistage sampling design to include a representative sample of about 10,000 individuals in each survey cycle. A detailed description of the NHANES database is publicly available (http://www.cdc.gov/nchs/nhanes.htm). Written and informed consent is the basis for the Institutional Review Committee of the National Center for Health Statistics

approving the NHANES program. This study involves no personal information. All private data are kept confidential throughout, preventing privacy breaches. It has no commercial interests. And as this study complies with the Declaration of Helsinki, ethical review can be waived.

The present research included the participants from five consecutive survey cycles (NHANES 2007–2008–2015–2016). First, participants were excluded if they were pregnant or uncertain about their pregnancy status during the examination(n = 715), or were under 18 years old (n = 19,864). Participants with missing data on serum uric acid and depression status were excluded (n = 4401). Ultimately, the research encompassed a total of 20,804 qualified participants.

## Data collection

The assessment of depression was conducted utilizing the Patient Health Questionnaire-9 (PHQ-9),a nine-item screening instrument that asks questions about the frequency of symptoms of depression over the past 2 weeks. Response categories which include "not at all," "several days," "more than half the days" and "nearly every day" were given a score ranging from 0 to 3. A total score was calculated ranging from 0 to 27. A score of 10 or higher has been well validated and is used to define depression.

To determine the concentration of serum uric acid in serum, plasma, or urine, a timed endpoint method was employed. This method involves the oxidation of uric acid by uricase, resulting in the formation of allatoin and hydrogen peroxide. Subsequently, the hydrogen peroxide reacts with 4-aminoantipyrine (4-AAP) and 3,5-dichloro-2-hydroxybenzene sulfonate (DCHBS) in the presence of peroxidase, yielding a colored product. The system measures the change in absorbance at a specific wavelength of 520 nm at regular intervals. It is worth noting that the change in absorbance directly correlates with the concentration of serum uric acid present in the sample.

The survey was conducted by well-trained interviewers through questionnaires to collect information on demographic characteristics, lifestyle factors, and current prescription drugs. The demographic characteristics included sex (men, women), age, ethnicity (Mexican American, other Hispanic, non-Hispanic white and non-Hispanic black), education (less than high school, high school or above), and annual household income (low income < $45,000, high income ≥ $45,000). The Global Physical Activity Questionnaire was utilized to estimate physical activity by inquiring about the subjects' physical activity intensity, duration, and frequency. The level of physical activity was then measured by calculating the total metabolic equivalent (MET) minutes per week. The physical examinations of NHANES are carried out in a mobile physical examination center, using standardized protocols and calibration equipment, The detailed information of data collection is available on the website (https://wwwn.cdc.gov/nchs/nhanes/analyticguidelines.aspx).

## Statistical analysis

During the analysis, we took into consideration the varying probabilities of selection, the complex sample design, as well as nonresponse and noncoverage by employing sample weights. We computed the weighted means along with their respective 95% CIs for continuous variables, and the percentages along with their 95% CIs for categorical variables of the chosen covariates. Standard errors were determined through Taylor series linearization. To delineate the non-linear relationships between serum uric acid and depression, we employed restricted cubic splines with three knots placed at the 5th, 50th, and 95th percentiles. A multivariable logistic regression model with weights was utilized to assess the relationships between serum

uric acid categories and depression status, taking into account appropriate adjustments for age, education level, physical activity level, smoking, drinking status, income, creatinine, race/ethnicity, body mass index, and albumin-to-creatinine ratio. A statistical significance level of P < 0.05 was deemed appropriate. All the statistical analysis was conducted using the survey package in R version 4.0.3 (http://www.r-project.org).

## Results

The analysis encompassed a total of 20,804 participants (i.e., PHQ-9≥5), including 10,400 men and 10,404 women,6.25% (650) male and 4.32% (449) female endorsing Depression. Males and females had an average age of 42 years. And it is obvious that participants with Depression tend to be smokers and drinkers. Additionally, they are more likely to have less education, lower income and lower physical activity levels, as shown in Table 1. For both sexes, participants have higher ACR and GGT levels, it shows that people with kidney disease increased vulnerability to depression (Table 1).

Generalized Additive Models reveal a linear relationship between serum uric acid level, depression, and gender. For women, regardless of whether the serum uric acid level rises or falls, the risk of depression increases. However, for men, serum uric acid level is not associated with depression (Fig 1).

It shows that among men, when using the tertile 1 (< 327.1µmol/L) as a reference point, the changes in the risk of depression across each subgroup are not prominent along with the elevation of uric acid levels. Among females, with the tertile 2 (249.8–303.3µmol/L) set as the reference. Among diabetic patients, The OR of tertile 1 (< 249.8 µmol/L) was 1.72 (95% *CI*:0.92–3.22), and the adjusted OR was 1.12 (95% *CI*: 0.30–4.90); the OR of tertile 3 (≥303.3 µmol/L) was 1.97 (95% *CI*:1.24–3.13), and the adjusted OR was 1.79 (95% *CI*:1.65–1.95). Among non-diabetic participants, the OR of tertile 1 (< 249.8 µmol/L) was 0.91 (95% *CI*:0.54–1.52), and the adjusted OR was 1.01 (95% *CI*:0.40–2.59); the OR of tertile 3 (≥303.3 µmol/L) was 1.10 (95% *CI*:0.81–1.50), and the adjusted OR was 0.95 (95% *CI*:0.71–1.27). Overall, an increase in uric acid levels is related to a higher risk of depression among female diabetes patients, yet this correlation is not so obvious among non-diabetic females (Table 2).

In male subjects, after adjusting for multiple factors, there is no significant correlation between different levels and classifications of uric acid and the symptoms of depression in the PHQ-9 questionnaire. However, among females, taking "Trouble concentrating on things (Q7)" as an example, in the diabetes group, the OR of tertile 1 (<249.8 µmol/L) was 1.85 (95% *CI*:0.55–6.28), and the OR of tertile 3 (≥303.3 µmol/L) was 2.28 (95% *CI*:1.13–4.60). There is a certain relationship between uric acid levels and specific symptoms of depression (Table 3, Table 4).

## Discussion

This national representative cross-sectional study has shown a link between serum uric acid level and depression, and gender differences. There is a paucity of studies examining the gender relationship between uric acid and depression. Thus, we stratified this relationship by gender to explore it and found meaningful results. According to this study, we discovered that in the American female group, a higher level of serum uric acid is correlated with a higher risk of depression, yet in the male group, the opposite is true.

Previous research has demonstrated that men typically possess higher serum uric acid levels compared with women. Intriguingly, they present a lower prevalence of depression, which is in accordance with our findings [13]. However, Yaru Li's study indicated a negative correlation between serum uric acid levels and the prevalence of depression among men, but has

**Table 1. The baseline characteristics of included subjects.**

| Variables | Endorsing depression symptoms(male/female) | Not endorsing depression symptoms(male/female) |
|---|---|---|
| N %(95%CI) | 6.25(5.56-6.94)/ 4.32(3.82-4.81) | 93.74(93.06-94.44)/ 95.68(95.19-96.18) |
| Uric acid(umol/L) | 358.05(350.58-365.51)/ 292.94(284.90-300.98) | 361.60(359.27-363.94)/ 283.04(281.25-284.83) |
| Race %(95%CI) | | |
| race1 | 9.71(6.93-12.49)/ 7.91(5.07-10.76) | 10.43(8.44-12.43)/ 8.59(6.89-10.29) |
| race2 | 8.10(5.79-10.40)/ 10.20(7.23-13.17) | 5.92(4.78-7.07)/ 5.82(4.65-6.32) |
| race3 | 62.89(57.33-68.43)/ 16.47(12.66-20.28) | 65.26(61.85-68.67)/ 60.48(54.10-66.86) |
| race4 | 12,70(9.50-15.92)/ 16.47(12.66-20.28) | 10.61(8.44-12.43)/ 12.24(10.31-14.17) |
| race5 | 6.60(4.37-8.82/ 4.93(2.43-7.43) | 7.78(6.67-8.89)/ 7.42(6.50-8.33) |
| Age,year(95%CI) | 42.20(40.95-43.45)/ 44.53(42.79-46.26) | 42.55(42.00-43.10)/ 45.41(44.79-46.03) |
| Education, %(95%CI) | | |
| education1 | 8.13(5.78-10.48)/ 8.96(6.07-11.85) | 5.21(4.45-5.96)/ 4.28(4.18-5.38) |
| education2 | 15.88(12.87-18.90)/ 21.82(17.06-26.58) | 11.47(10.22-12.72)/ 9.93(8.85-11.00) |
| education3 | 28.30(23.55-33.05)/ 23,64(18.22-29.06) | 23.05(21.65-24.44)/ 20.89(19.56-22.21) |
| education4 | 32.95(26.90-39.00)/ 33.02(26.83-39.21) | 29.75(28.56-30.93)/ 33.73(32.27-35.20) |
| education5 | 14.74(9.52-19.95)/ 12.57(8.51-16.63) | 30.46(28.20-32.71)/ 30.63(28.24-33.02) |
| Income (95%CI) | 9.87(8.67-10.98)/ 8.96(7.68-10.25) | 11.94(11.53-12.35)/ 11.48(11.06-11.90) |
| Smoking,% (95%CI) | | |
| yes | 63.88(58.93-68.82)/ 59.79(54.85-64.74) | 47.38(45.73-49.02)/ 35.62(33.99-37.25) |
| no | 36.12(31.18-41.06)/ 40.21(35.26-45.15) | 52.58(50.94-54.23)/ 64.31(62.70-65.92) |
| Drinking,%(95%CI) | | |
| yes | 83.27(79.54-87.00)/ 70.36(65.29-75.42) | 84.68(83.89-86.42)/ 68.21(65.91-70.52) |
| no | 16.60(12.90-20.30)/ 29.08(24.23-33.93) | 15.24(13.51-16.98)/ 31.74(29.43-34.05) |
| Met-activity(95%CI) | 5650.216(3101.264-8199.169)/ 2257.20(1763.15-2751.26) | 5841.14(5406.086-6276.194)/ 2959.90(2670.36-3249.44) |
| Systolic blood pressure (mmHg) (95%CI) | 123.96(121.93-125.99)/ 119.79(117.75-121.84) | 122.65(122.16-123.13)/ 119.11(118.59-119.63) |
| Diastolic blood pressure (mmHg) (95%CI) | 73.08(71.94-74.21)/ 70.17(68.65-71.68) | 71.63(71.04-72.22)/ 68.78(68.26-69.30) |
| HbA1c(%) (95%CI) | 5.63(5.55-5.71)/ 5.71(5.61-5.81) | 5.58(5.55-5.61)/ 5.56(5.53-5.58) |
| Plasma glucose:(mmol/L) (95%CI) | 6.04(5.83-6.26)/ 6.06 | 5.98(5.92-6.04)/ 5.80-6.32) |
| Insulin (uU/mL) (95%CI) | 15.06(13.24-16.88)/ 15.45(13.71-17.20) | 13.66(12.99-14.33)/ 12.11(11.71-12.51) |
| Ogtt(mmol/L) (95%CI) | 6.37(5.93-6.81)/ 6.84(6.34-7.34) | 6.18(6.07-6.29)/ 6.43(6.32-6.53) |
| TG (mg/dL) (95%CI) | 1.85(1.63-2.07)/ 1.49(1.34-1.65) | 1.52(1.47-1.58)/ 1.26(1.22-1.30) |
| LDL (mg/dL) (95%CI) | 3.01(2.90-3.13)/ 3.06(2.86-3.25) | 2.98(2.94-3.01)/ 2.95(2.91-2.98) |
| AST (u/L) (95%CI) | 28.39(26.68-30.09)/ 26,59(22.87-30.30) | 27.70(27.35-28.09)/ 27.70(27.35-28.09) |
| ALT (u/L) (95%CI) | 31.27(29.26-33.27)/ 23.95(21.91-25.99) | 30.22(29.67-30.77)/ 21.04(20.71-21.37) |
| GGT(u/L) (95%CI) | 38.70(34.16-43.24)/ 32.65(27.32-37.97) | 30.56(29.76-31.36)/ 22.29(21.24-23.33) |
| Cr | 0.98(0.95-1.01)/ 0.80(0.76-0.84) | 0.97(0.97-0.98)/ 0.76(0.76-0.77) |
| Acr | 45.81(26.86-64.76)/ 52.06(18.16-85.97) | 26.05(20.75-31.34)/ 27.22(23.04-31.40) |
| PrediagDM2, % (95%CI) | | |
| yes | 7.98(6.11-9.86)/ 13.97(10.59-17.34) | 7.21(6.53-7.88)/ 7.16(6.46-7.87) |
| no | 88.00(85.39-90.61)/ 82.18(77.97-86.39) | 91.12(90.40-91.83)/ 90.99(90.22-91.75) |

This table presents the baseline characteristics of the study population, stratified by the presence or absence of depression symptoms and by gender (male/female).The data are presented as means with 95% confidence intervals or percentages with 95% confidence intervals.

nothing to do with the prevalence of depression in women [10], and Meng's analysis showed that depressed patients have lower serum uric acid compared to normal population whether man or woman [12]. In addition, a Korean survey agrees with our findings, reporting that the prevalence of depressive symptoms was elevated in women with lower serum uric acid levels

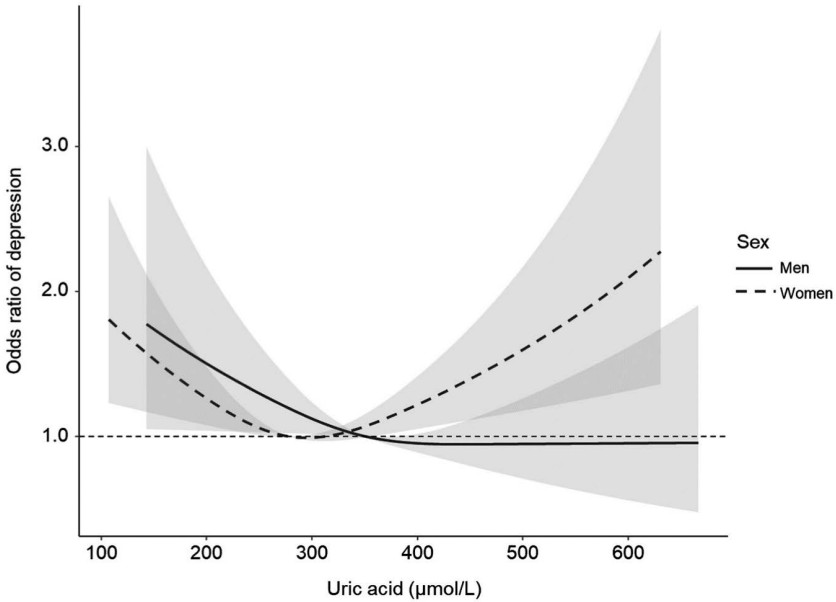

**Fig 1. Adjusted cubic splines for the odds ratio of depression by uric acid levels in men and women.** This figure presents adjusted cubic spline curves illustrating the relationship between serum uric acid levels and the odds ratio (OR) of depression in men and women. Higher uric acid levels may be associated with lower depression risk in men, while higher levels may be associated with increased risk in women. Model was adjusted for age, education level, physical activity level, smoking status, drinking status, income, creatinine, race/ethnicity, body mass index, and albumin-to-creatinine ratio.

**Table 2. The association between uric acid categories and depression by gender and diabetes status.**

| Uric acid category | Diabetes | | | Non-diabetes | | | Total | | |
|---|---|---|---|---|---|---|---|---|---|
| | Case/ total | Unadjusted OR (95%CI) | Adjusted OR (95%CI) | Case/ total | Unadjusted OR (95%CI) | Adjusted OR (95%CI) | Case/ total | Unadjusted OR (95%CI) | Adjusted OR (95%CI) |
| **Men** | | | | | | | | | |
| Tertile 1 ( < 327.1 μmol/L) | 40/ 553 | 1.00 (Ref) | 1.00 (Ref) | 188/ 2904 | 1.00 (Ref) | 1.00 (Ref) | 228/ 3464 | 1.00 (Ref) | 1.00 (Ref) |
| Tertile 2 (327.1 ~ 386.6 μmol/L) | 26/ 400 | 0.62 (0.29-1.32) | 0.50 (0.19-1.31) | 177/ 3228 | 0.88 (0.66-1.18) | 0.91 (0.62-1.33) | 203/ 3631 | 0.84 (0.63-1.11) | 0.84 (0.58-1.22) |
| Tertile 3 ( ≥ 386.6 μmol/L) | 33/ 379 | 0.89 (0.49-1.64) | 0.72 (0.36-1.42) | 158/ 2923 | 0.95 (0.73-1.25) | 0.95 (0.64-1.41) | 191/ 3305 | 0.93 (0.73-1.19) | 0.87 (0.62-1.22) |
| **Women** | | | | | | | | | |
| Tertile 1 ( < 249.8 μmol/L) | 45/ 331 | 1.72 (0.92-3.22) | 1.12 (0.30-4.90) | 312/ 3202 | 0.91 (0.54-1.52) | 1.01 (0.40-2.59) | 358/ 3538 | 0.95 (0.54-1.67) | 1.04 (0.39-2.78) |
| Tertile 2 (249.8 ~ 303.3 μmol/L) | 48/ 262 | 1.00 (Ref) | 1.00 (Ref) | 289/ 3037 | 1.00 (Ref) | 1.00 (Ref) | 337/ 3306 | 1.00 (Ref) | 1.00 (Ref) |
| Tertile 3 ( ≥ 303.3 μmol/L) | 126/ 717 | 1.97 (1.24-3.13) | 1.79 (1.65-1.95) | 289/ 2835 | 1.10 (0.81-1.50) | 0.95 (0.71-1.27) | 418/ 3560 | 1.27 (1.00-1.61) | 1.05 (0.92-1.19) |

This table presents the association between uric acid levels (categorized into tertiles) and depression, stratified by gender and diabetes status. In men, higher uric acid levels are inversely associated with depression risk, while in women, higher uric acid levels are positively associated with depression risk, especially among those with diabetes.

than in those with higher serum uric acid levels and it is no significant associations between serum uric acid levels and depressive symptoms was observed in men [14]. The reasons for this may be that we use different depression scales, there are significant variations in the

**Table 3. The odds ratios of the associations between uric acid categories and each item in PHQ-9 diabetes status in men with tertile 1 of uric acid (<327.1 µmol/L) as reference.**

| | Diabetes | | Non-diabetes | | Total | |
|---|---|---|---|---|---|---|
| | Tertile 2 (327.1 ~ 386.6 µmol/L) | Tertile 3 (≥386.6 µmol/L) | Tertile 2 (327.1 ~ 386.6 µmol/L) | Tertile 3 (≥386.6 µmol/L) | Tertile 2 (327.1 ~ 386.6 µmol/L) | Tertile 3 (≥386.6 µmol/L) |
| **Q1: Have little interest in doing things** | 0.49 (0.25-0.94) | 0.76 (0.31-1.88) | 0.74 (0.54-1.02) | 0.77 (0.58-1.03) | 0.71 (0.52-0.97) | 0.75 (0.57-1.00) |
| **Q2: Feeling down, depressed, or hopeless** | 0.65 (0.28-1.53) | 0.69 (0.32-1.49) | 0.72 (0.51-1.06) | 0.83 (0.57-1.21) | 0.73 (0.51-1.05) | 0.80 (0.56-1.14) |
| **Q3: Trouble sleeping or sleeping too much** | 0.44 (0.26-0.77) | 0.98 (0.53-1.81) | 0.76 (0.56-1.04) | 0.95 (0.70-1.28) | 0.72 (0.53-0.96) | 0.93 (0.72-1.21) |
| **Q4: Feeling tired or having little energy** | 0.91 (0.48-1.70) | 1.20 (0.69-2.07) | 0.88 (0.67-1.16) | 0.86 (0.64-1.15) | 0.88 (0.68-1.14) | 0.86 (0.66-1.12) |
| **Q5: Poor appetite or overeating** | 0.38 (0.16-0.90) | 0.56 (0.29-1.08) | 0.92 (0.67-1.26) | 1.09 (0.79-1.49) | 0.81 (0.62-1.07) | 0.97 (0.75-1.26) |
| **Q6: Feeling bad about yourself** | 0.34 (0.12-0.95) | 0.38 (0.14-1.01) | 0.95 (0.67-1.34) | 0.85 (0.58-1.25) | 0.88 (0.64-1.21) | 0.78 (0.56-1.09) |
| **Q7: Trouble concentrating on things** | 0.26 (0.09-0.71) | 0.52 (0.18-1.45) | 0.85 (0.57-1.27) | 0.89 (0.61-1.29) | 0.75 (0.51-1.11) | 0.79 (0.53-1.18) |
| **Q8: Moving or speaking slowly or too fast** | 0.90 (0.33-2.45) | 0.62 (0.28-1.34) | 0.72 (0.45-1.15) | 0.72 (0.43-1.22) | 0.75 (0.51-1.09) | 0.69 (0.45-1.05) |
| **Q9: Thought you would be better off dead** | 0.33 (0.00-46.40) | 0.06 (0.00-2.65) | 0.92 (0.67-1.26) | 0.76 (0.20-2.81) | 0.83 (0.39-1.77) | 0.63 (0.22-1.81) |

This table presents the odds ratios (OR) and 95% confidence intervals (CI) for the association between uric acid categories (Tertile 2 and Tertile 3) and each item of the Patient Health Questionnaire-9 (PHQ-9) in men, stratified by diabetes status (with and without diabetes). The analysis uses the lowest uric acid tertile (<327.1 µmol/L) as the reference group. The results suggest that higher uric acid levels may have differential associations with specific depressive symptoms, particularly among men with diabetes.

**Table 4. The odds ratios of the associations between uric acid categories and each item in PHQ-9 diabetes status in women with tertile 2 of uric acid (249.8 ~ 303.3 µmol/L) as reference.**

| | Diabetes | | Non-diabetes | | Total | |
|---|---|---|---|---|---|---|
| | Tertile 1 (<249.8 µmol/L) | Tertile 3 (>=303.3 µmol/L) | Tertile 1 (<249.8 µmol/L) | Tertile 3 (>=303.3 µmol/L) | Tertile 1 (<249.8 µmol/L) | Tertile 3 (>=303.3 µmol/L) |
| **Q1: Have little interest in doing things** | 1.22 (0.19-7.75) | 1.30 (0.84-2.01) | 1.18 (0.80-1.75) | 1.04 (0.36-2.99) | 1.21 (0.77-1.92) | 1.08 (0.55-2.13) |
| **Q2: Feeling down, depressed, or hopeless** | 1.52 (0.11-20.28) | 1.29 (0.50-3.34) | 1.09 (0.64-1.86) | 0.94 (0.41-2.15) | 1.13 (0.53-2.39) | 0.99 (0.47-2.06) |
| **Q3: Trouble sleeping or sleeping too much** | 0.76 (0.41-1.39) | 1.30 (0.66-2.55) | 1.12 (0.36-3.44) | 0.97 (0.50-1.88) | 1.11 (0.39-3.19) | 1.02 (0.66-1.58) |
| **Q4: Feeling tired or having little energy** | 0.82 (0.34-1.99) | 1.29 (1.06-1.55) | 0.90 (0.36-2.24) | 0.99 (0.95-1.03) | 0.91 (0.37-2.22) | 1.05 (1.00-1.09) |
| **Q5: Poor appetite or overeating** | 0.92 (0.41-2.03) | 1.32 (0.69-2.52) | 0.99 (0.27-3.63) | 0.87 (0.46-1.64) | 0.99 (0.28-3.47) | 0.92 (0.58-1.46) |
| **Q6: Feeling bad about yourself** | 1.24 (0.95-1.61) | 1.12 (0.31-4.03) | 0.93 (0.53-1.61) | 1.04 (1.02-1.06) | 0.96 (0.58-1.58) | 1.05 (0.89-1.25) |
| **Q7: Trouble concentrating on things** | 1.85 (0.55-6.28) | 2.28 (1.13-4.60) | 0.97 (0.85-1.11) | 1.01 (0.46-2.23) | 1.03 (0.85-1.26) | 1.11 (0.57-2.15) |
| **Q8: Moving or speaking slowly or too fast** | 1.13 (0.32-4.08) | 1.02 (0.95-1.10) | 1.05 (0.88-1.27) | 1.24 (0.35-4.37) | 1.04 (0.76-1.42) | 1.19 (0.42-3.35) |
| **Q9: Thought you would be better off dead** | 0.40 (0.21-0.74) | 0.64 (0.12-3.34) | 1.02 (0.26-3.96) | 1.55 (0.95-2.52) | 0.89 (0.17-4.58) | 1.35 (0.56-3.25) |

This table presents the odds ratios (OR) and 95% confidence intervals (CI) for the association between uric acid categories (Tertile 1 and Tertile 3) and each item of the Patient Health Questionnaire-9 (PHQ-9) in women, stratified by diabetes status (with and without diabetes). The analysis uses the tertile 2 of uric acid (249.8 ~ 303.3 µmol/L) as the reference group. The results suggest that both lower and higher uric acid levels may have differential associations with specific depressive symptoms, particularly among women with diabetes.

diagnostic criteria for depression among different classifications. Additionally, differences in race and lifestyle may also contribute to these varying results.

The mechanism of serum uric acid and depression is not clear. Uric acid is an antioxidant, and has different physiological effects in the occurrence and development of different diseases. Under normal physiological conditions, serum uric acid can interact with a variety of oxidants, including hydrogen peroxide and hydroxyl radicals, to clear reactive oxygen species (ROS), inhibit lipid peroxidation, avoid oxidative damage, and effectively play a neuroprotective role [15,16]. However, other study shows that low serum uric acid level in central nervous system may damage the antioxidant capacity of cells [17]. Levels of uric acid and purines, such as adenosine, play a role in regulating mood, sleep, activity, appetite, cognition,

memory, susceptibility to seizures, social interaction, and impulsivity [18]. The inhibition of A1 receptor and the activation of A2A receptor are influenced by purine, and purinergic signaling plays a pivotal role in the pathophysiology of depression [19]. Serum uric acid is the final product of endogenous purine metabolism, maybe also play an important role. Luca's study showed that both inflammation and oxidative stress have been shown to cause vascular endothelial damage, which is implicated in the occurrence and progression of various diseases, including vascular depression and dementia [20]. The level of uric acid depends on the dynamic balance between the intake of purine-rich foods, endogenous uric acid synthesis, and excretion of uric acid through various pathways (including urine and the gastrointestinal tract) [21,22]. Studies show that age, diet, alcohol consumption, fructose-rich intake, drug intervention, and diseases (including obesity, insulin resistance and kidney diseases) can all contribute to the development of hyperuricemia [23–25]. Therefore, in order to reduce the influence of other variables, the statistical model of this study has adjusted for age, education level, physical activity level, smoking status, drinking status, income, creatinine, race/ethnicity, body mass index, and albumin-to-creatinine ratio.

This study has several strengths. First, there is a paucity of research examining the gender-specific connection between serum uric acid and depression. Secondly, a scarcity of studies has delved into the interactions of serum uric acid with various factors, including age, education level, activity, smoking, drinking, income, creatinine, race/ethnicity, body mass index, and albumin-to-creatinine ratio, in relation to depression prevalence. Thirdly, the study utilizes the NHANES dataset, which employs a complex, multi-stage probabilistic sampling design, thus enhancing the significance of our findings. However, there are also some limitations to consider. Firstly, the cross-sectional nature of the study precludes any definitive conclusions regarding causality between serum uric acid and depression. Secondly, the individuals who were investigated were exclusively middle-aged and senior Americans, limiting the generalizability of these findings to other populations. Finally, depression was assessed using the PHQ-9 questionnaire, which may introduce potential inaccuracies.

## Conclusion

In summary, this study reveals a significant gender-specific association between serum uric acid levels and depression in U.S. adults. Our findings indicate that higher serum uric acid levels are inversely associated with depression in men, suggesting a potential protective role against depressive symptoms. Conversely, in women, elevated serum uric acid levels are identified as an independent risk factor for depression, particularly among those with diabetes. This highlights the importance of gender as a critical factor in understanding the complex relationship between uric acid and depression. Moreover, management of uric acid levels is crucial for individuals with depression and diabetes. Further exploration into the mechanistic links between uric acid, depression, and gender is warranted to inform more effective therapeutic strategies.

## Acknowledgements

We thank the staff of the National Center for Health Statistics for the research data it has provided and anonymous reviewers for valuable feedback on a previous version of this manuscript.

## Author contributions

**Conceptualization:** Ertao Zhang.

**Data curation:** Zhao Dong, Longyu Wang, Can Zhao.

**Formal analysis:** Ertao Zhang, Jie Li, Yan Liu, Zhao Dong, Longyu Wang, Can Zhao.

**Funding acquisition:** Jie Li, Yan Liu, Zhao Dong.

**Investigation:** Ertao Zhang, Jie Li, Yan Liu, Zhao Dong, Can Zhao.

**Methodology:** Ertao Zhang, Longyu Wang, Can Zhao.

**Project administration:** Ertao Zhang, Jie Li.

**Resources:** Yan Liu, Can Zhao.

**Software:** Longyu Wang, Can Zhao.

**Supervision:** Ertao Zhang, Jie Li, Yan Liu, Zhao Dong.

**Visualization:** Yan Liu.

**Writing – original draft:** Jie Li.

**Writing – review & editing:** Ertao Zhang, Jie Li, Yan Liu.

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
