## [Decision Letter · Decision Letter 0]

5 Nov 2024

PONE-D-24-31772Associations between serum uric acid and depression in U.S. adults: an analysis of the National Health and Nutrition Examination Survey 2007–2016PLOS ONE

Dear Dr. ertao, 

Thank you for submitting your manuscript to PLOS ONE. After careful consideration, we feel that it has merit but does not fully meet PLOS ONE’s publication criteria as it currently stands. Therefore, we invite you to submit a revised version of the manuscript that addresses the points raised during the review process.

Comments: In general, the article is well written. Please add more discussions to reveal the association between gender disparities, serum uric acid level and depression.

We look forward to receiving your revised manuscript.

Kind regards,

Lai Kuan Lee

Academic Editor

PLOS ONE

Journal Requirements:

4. Please ensure that you include a title page within your main document. You should list all authors and all affiliations as per our author instructions and clearly indicate the corresponding author.

5. Please remove your figures from within your manuscript file, leaving only the individual TIFF/EPS image files, uploaded separately. These will be automatically included in the reviewers’ PDF.

**Additional Editor Comments:**

The research article entitled "Associations between serum uric acid and depression in U.S. adults: an analysis of the National Health and Nutrition Examination Survey 2007–2016" is reviewed.

In general, the article is important and add literature to the existing pool.

More discussions are expected to reveal the connection between gender, serum uric acid and depression.

Reviewers' comments:

Reviewer's Responses to Questions

**Comments to the Author**

1. Is the manuscript technically sound, and do the data support the conclusions?

Reviewer #1: Yes

Reviewer #2: Yes

2. Has the statistical analysis been performed appropriately and rigorously? 

Reviewer #1: Yes

Reviewer #2: Yes

3. Have the authors made all data underlying the findings in their manuscript fully available?

Reviewer #1: Yes

Reviewer #2: Yes

4. Is the manuscript presented in an intelligible fashion and written in standard English?

Reviewer #1: Yes

Reviewer #2: Yes

5. Review Comments to the Author

Reviewer #1: Hello. Thanks for opportunity to review this manuscript.

Your contribution to the already available research on this topic is to add that there is a gender difference in the relationship between uric acid and depression.

- Your citation of reference # 11 was only done on male adolescents. Please change the language you have used to seem to indicate the relationship does not exist in women when what you wanted to convey was that there was no data on women.

- In results, The paragraph starting with "As shown in Figure 1,..." could use a change in grammar and syntax to better explain the chart. (e.g- whether serum uric acid goes up or down, it increases the risk of depression in women"). The relationship between different tertiles and the depression incidence could be explained better as well. There is a typo of "Man" for "men" as well. There would be a better way of explaining the general relationship and the one when diabetes is involved separately.

- The same in the Discussion section- "Higher serum uric acid has a correlation with a higher risk of depression in American female population, but it's opposite in male". That's partially true but can be described in a better way.

- Next paragraph in Discussion has typo- Why is it surprising that men show lower prevalence of depression with higher level of serum uric acid (isn't that what previous data has shown?) -- also "it's similar to our found" is a typo.

- The conclusion might also need to be rewritten with correction of typos ("uric acid is independent risk factors", "people with diabetes also increase the risk of depression..."). Also, adding diabetes as a risk factor for depression seems unrelated to your study about uric acid. If it is not, please explain it.

- Then in Discussion you mention other studies that show results opposite or not in keeping with your results which brings up a big question about the veracity and reproducibility of these results. There are many confounding factors that can lead to uric acid increase over the ones related to depression and I think the connection between the two especailly in a cross sectional data does not explain much. Please discuss other confounders in discussion. What other factors can determine uric acid levels? What other biochemical reactions cause uric acid to be formed other than purines?

Reviewer #2: abstract keywords ; should we put?..... oxidative stress; diabetes

Introduction: 6-7th line and 16-17th: repeated "serum uric acid contributes

to approximately 60% of the antioxidant capacity in body's blood"

Methods and material:

Study design and subjects: last line: do 20804 participants have data on uric acid and depression? as per the statement

Result: 3rd line- rephrase....

symptoms and illness are different. It is better to mention only : Depression

Was 19826 Participants score of PHQ less than 5?

minor grammatical issues: highlighted pink

6. PLOS authors have the option to publish the peer review history of their article (what does this mean?). If published, this will include your full peer review and any attached files.

Reviewer #1: No

Reviewer #2: No

---

## [Author Response · Author response to Decision Letter 1]

4 Dec 2024

I'm extremely grateful for the comments from all the reviewers.

---

## [Editor Report · Decision Letter 1]

30 Dec 2024

PONE-D-24-31772R1Associations between serum uric acid and depression in U.S. adults: an analysis of the National Health and Nutrition Examination Survey 2007–2016PLOS ONE

Dear Dr. li,

Thank you for submitting your manuscript to PLOS ONE. After careful consideration, we feel that it has merit but does not fully meet PLOS ONE’s publication criteria as it currently stands. Therefore, we invite you to submit a revised version of the manuscript that addresses the points raised during the review process.

The manuscript entitled "Associations between serum uric acid and depression in U.S. adults: an analysis of the National Health and Nutrition Examination Survey 2007–2016" is reviewed again.

Few comments:

- The language and grammar mistakes in the manuscript require further polishing.

Eg: The study show that 

       it as the independent 

- Table 2 is being presented in a not readable format.

- Please revise the authors' contributions. 

-Ethical approval: Was the publication consent granted? Please list the human ethical approval serial number/reference code.

- Add limitation and strengths

We look forward to receiving your revised manuscript.

Kind regards,

Lai Kuan Lee

Academic Editor

PLOS ONE

Additional Editor Comments (if provided):

The manuscript entitled "Associations between serum uric acid and depression in U.S. adults: an analysis of the National Health and Nutrition Examination Survey 2007–2016" is reviewed again.

Few comments:

- The language and grammar mistakes in the manuscript require further polishing.

Eg: The study show that

it as the independent

- Table 2 is being presented in a not readable format.

- Please revise the authors' contributions.

-Ethical approval: Was the publication consent granted? Please list the human ethical approval serial number/reference code.

- Add limitation and strengths

---

## [Editor Report · Decision Letter 2]

5 Feb 2025

PONE-D-24-31772R2Associations between serum uric acid and depression in U.S. adults: an analysis of the National Health and Nutrition Examination Survey 2007–2016PLOS ONE

Dear Dr. li,

Thank you for submitting your manuscript to PLOS ONE. After careful consideration, we feel that it has merit but does not fully meet PLOS ONE’s publication criteria as it currently stands. Therefore, we invite you to submit a revised version of the manuscript that addresses the points raised during the review process.

**Dear authors,**

**The revised manuscript is thoroughly checked. **

**Some comments: - The language and grammar mistakes in the manuscript still exist. **

**You are advised to check every single information in the text, including the reference list. Example: Line 20, Line 27-28, Line 200, Line 203 **

**- Please provide ethical approval reference number. **

**- Provide descriptive footnote for every tables. - Arrange Table 2 and Table 3 in landscape mode. **

**- Line 220: Write a meaningful acknowledgement **

**- Add clear objectives, hypothesis**

We look forward to receiving your revised manuscript.

Kind regards,

Lai Kuan Lee

Academic Editor

PLOS ONE

**Additional Editor Comments:**

Dear authors,

The revised manuscript is thoroughly checked.

Some comments:

- The language and grammar mistakes in the manuscript still exist. You are advised to check every single information in the text, including the reference list.

Example: Line 20, Line 27-28, Line 200, Line 203

- Please provide ethical approval reference number.

- Provide descriptive footnote for every tables.

- Arrange Table 2 and Table 3 in landscape mode.

- Line 220: Write a meaningful acknowledgement

- Add clear objectives, hypothesis

---

## [Author Response · Author response to Decision Letter 3]

17 Feb 2025

Thank you very much for your suggestions. They have been extremely helpful to us.

---

## [Editor Report · Decision Letter 3]

5 Mar 2025

Associations between serum uric acid and depression in U.S. adults: an analysis of the National Health and Nutrition Examination Survey 2007–2016

PONE-D-24-31772R3

Dear Dr. Jie Li,

We’re pleased to inform you that your manuscript has been judged scientifically suitable for publication and will be formally accepted for publication once it meets all outstanding technical requirements.

Kind regards,

Lai Kuan Lee

Academic Editor

PLOS ONE
---

## [Editor Report · Acceptance letter]

PONE-D-24-31772R3

PLOS ONE

Dear Dr. Li,

I'm pleased to inform you that your manuscript has been deemed suitable for publication in PLOS ONE. Congratulations! Your manuscript is now being handed over to our production team.

Kind regards,

on behalf of

Dr. Lai Kuan Lee

Academic Editor

PLOS ONE